# *Vernonia polyanthes* Less. (Asteraceae Bercht. & Presl), a Natural Source of Bioactive Compounds with Antibiotic Effect against Multidrug-Resistant *Staphylococcus aureus*

**DOI:** 10.3390/antibiotics12030622

**Published:** 2023-03-21

**Authors:** Jordana Damasceno Gitirana de Santana, Oscar Alejandro Santos-Mayorga, Jônatas Rodrigues Florencio, Mirella Chrispim Cerqueira de Oliveira, Luísa Maria Silveira de Almeida, Julianna Oliveira de Lucas Xavier, Danielle Cristina Zimmermann-Franco, Gilson Costa Macedo, Adriana Lúcia Pires Ferreira, Orlando Vieira de Sousa, Ademar Alves da Silva Filho, Maria Silvana Alves

**Affiliations:** 1Programa de Pós-Graduação em Ciências Farmacêuticas, Universidade Federal de Juiz de Fora, Rua José Lourenço Kelmer s/n, Campus Universitário, Bairro São Pedro, Juiz de Fora 36036-900, MG, Brazil; 2Centro de Tecnologia Celular e Imunologia Aplicada (IMUNOCET), Departamento de Parasitologia, Microbiologia e Imunologia, Instituto de Ciências Biológicas, Universidade Federal de Juiz de Fora, Rua José Lourenço Kelmer s/n, Campus Universitário, Bairro São Pedro, Juiz de Fora 36036-900, MG, Brazil; 3Hospital Universitário Clementino Fraga Filho, Universidade Federal do Rio de Janeiro, Rua Professor Rodolpho Paulo Rocco 255, Cidade Universitária, Ilha do Fundão, Rio de Janeiro 21941-913, RJ, Brazil; 4Departamento de Ciências Farmacêuticas, Faculdade de Farmácia, Universidade Federal de Juiz de Fora, Rua José Lourenço Kelmer s/n, Campus Universitário, Bairro São Pedro, Juiz de Fora 36036-900, MG, Brazil

**Keywords:** *Vernonia polyanthes*, flavonoids, glaucolide A, UHPLC/Q-TOF-MS, antibacterial activity, methicillin-resistant *Staphylococcus aureus*

## Abstract

*Vernonia polyanthes* is a medicinal plant used to treat many disorders, including infectious diseases. This study investigated the chemical constituents and the antibacterial activity of *V. polyanthes* leaf rinse extract (Vp-LRE). The chemical characterization of Vp-LRE was established using ultra-high performance liquid chromatography coupled to quadrupole time-of-flight mass spectrometry (UHPLC/Q-TOF-MS), and glaucolide A was identified through ^1^H and ^13^C nuclear magnetic resonance (NMR) and mass fragmentation. The cytotoxicity was evaluated using 3-(4,5-dimethylthiazol-2-yl)-2,5-diphenyl tetrazolium bromide (MTT). The antibacterial activity was assessed by minimal inhibitory concentration and minimal bactericidal concentration. Interactions between ligands and beta-lactamase were evaluated via molecular docking. UHPLC/Q-TOF-MS detected acacetin, apigenin, chrysoeriol, isorhamnetin, isorhamnetin isomer, kaempferide, 3′,4′-dimethoxyluteolin, 3,7-dimethoxy-5,3′,4′-trihydroxyflavone, piptocarphin A and glaucolide A. Vp-LRE (30 µg/mL) and glaucolide A (10 and 20 μg/mL) were cytotoxic against RAW 264.7 cells. Glaucolide A was not active, but Vp-LRE inhibited the *Staphylococcus aureus*, methicillin-resistant *S. aureus* (MRSA), *Escherichia coli*, *Salmonella* Choleraesuis and Typhimurium, with a bacteriostatic effect. The compounds (glaucolide A, 3′,4′-dimethoxyluteolin, acacetin and apigenin) were able to interact with beta-lactamase, mainly through hydrogen bonding, with free energy between −6.2 to −7.5 kcal/mol. These results indicate that *V. polyanthes* is a potential natural source of phytochemicals with a significant antibiotic effect against MRSA strains.

## 1. Introduction

Infectious diseases are a major threat to human health and are responsible for high rates of morbidity and mortality. The treatment of these diseases is conducted with antibiotics that are often used irrationally, causing the emergence of microbial resistance (in addition to adverse effects) [1]. The declining effectiveness of antibiotics in treating bacterial infections worldwide is evident. The antimicrobial resistance in pathogenic bacteria has become one of the most serious and complex public health problems of the 21st century [2] with worrying clinical and economic consequences for sanitary and health agencies [3].

In function of this alarming scenario, the World Health Organization (WHO) published the top 20 drug-resistant bacteria list at a global level, with critical (priority 1), high (priority 2) and medium (priority 3) priorities, requiring prompt action from the scientific research community to develop new antibiotics. *Escherichia coli* (carbapenem-resistant, 3rd generation cephalosporin-resistant) (priority 1), *Pseudomonas aeruginosa* (carbapenem resistant) (priority 1), *Staphylococcus aureus* [methicillin-resistant (MRSA), vancomycin intermediate (VISA) and resistant (VRSA)] (priority 2) and *Salmonella* spp. (fluoroquinolone-resistant) (priority 2) strains are major concerns [4]. In this sense, the research of new therapeutic agents represents a relevant strategy in the combat of infections caused by multidrug-resistant bacteria, including MRSA.

The antimicrobial properties of compounds produced by plants have been recognized since antiquity and have been scientifically investigated in the last decades, and several natural products are recognized as antibacterial agents [5,6,7,8]. This scientific framework associated with the development of new technologies justifies the renewed interest in the screening of natural products to discovery of new substances with antibacterial potential. Even with the lack of support from major pharmaceutical industries, natural products still show a substantial impact on the remedy discovery process [8]. Nowadays, among all substances approved as new antibacterial chemical entities, a significant percentage of them are natural compounds or their synthetic derivatives [7,8].

*Vernonia polyanthes* Less. (Asteraceae Bercht. & Presl family), commonly known in Brazil as “assa-peixe”, occurs in South America and has been traditionally used mainly to treat infectious and inflammatory processes, as well as wounds, burns, respiratory system disorders, kidney dysfunctions and in an antimycobacterial capacity [9,10,11]. The therapeutic potential of *V. polyanthes* has been investigated, and its pharmacological activities, such as antihypertensive and diuretic [12], antileishmanial [13], antimycobacterial [11], antiulcerogenic [14], antinociceptive and anti-inflammatory [15], cytotoxic [16] and topical anti-inflammatory [17] effects have been reported. Although some reports about the antibacterial activities of *V. polyanthes* have been published [11,18,19,20], more investigations are necessary to establish the real antibacterial potential of this plant species, focusing on new antibiotics and new therapeutic approaches mainly to treat “untreatable” infectious diseases. Thus, in function of the medical importance in the last decades, it is relevant to research the antibacterial activity of *V. polyanthes* against multidrug-resistant clinical bacterial strains.

Based on the use of *V. polyanthes* leaves for the treatment of bacterial infectious diseases and the clinical relevance of resistant pathogens, the aim of this study was to evaluate the antibacterial activities of the leaf rinse extract of *V. polyanthes* (Vp-LRE) and its main compound glaucolide A against reference and clinical multidrug-resistant Gram-positive and Gram-negative bacterial strains.

## 2. Results

### 2.1. Production of the Extract of V. polyanthes

A crude extract of *V. polyanthes* was prepared by rinsing the leaves with dichloromethane. The rinsed leaves extract of *V. polyanthes* (Vp-LRE) was chosen for analysis since many bioactive compounds in *Vernonia* species (such as flavonoids and sesquiterpene lactones) are localized mainly at the glandular trichomes leaves [21,22,23].

In this present study, Vp-LRE was prepared and its antibacterial activity against ATCC^®^ and clinical multidrug-resistant Gram-positive and Gram-negative bacterial strains was evaluated, which have not been reported in the literature.

### 2.2. Total Phenolic and Flavonoid Content Determinations

The total phenolic content of Vp-LRE was spectrophotometrically determined using Folin-Ciocalteu reagent, expressed as gallic acid equivalents (g GAE/100 g) and calculated from the calibrated curve (r^2^ = 0.9970), while total flavonoid content was estimated using the aluminum chloride method and expressed as rutin equivalents (g RE/100 g) (r^2^ = 0.9995). In *V. polyanthes*, the values were 2.53 ± 0.01 g GAE/100 g (Vp-LRE) for phenols and 4.26 ± 0.04 g RE/100 g (Vp-LRE) for flavonoids.

### 2.3. Evaluation of the Chemical Composition of Vp-LRE by Ultra-High Performance Liquid Chromatography Coupled with Quadrupole Time-of-Flight Mass Spectrometry (UHPLC-MS-QTOF) Analysis

UHPLC-MS-QTOF analysis was applied for the qualitative chemical characterization of different metabolites in Vp-LRE. The components of Vp-LRE were separated and analyzed by UHPLC-MS-QTOF, and a typical UHPLC-MS-QTOF chromatogram of Vp-LRE, at negative ion mode, is shown in Figure 1. Glaucolide A and apigenin were identified and eight compounds were annotated, and detailed information of each peak is listed in Table 1. Peak annotation was based on each compound’s molecular ions and its MS/MS studies (Table 1), as well as comparison with the available literature. The options with the lower mass error values and the higher MS scores useful for obtaining better confident formula assignments were preferable. The structures of the proposed compounds were tentatively annotated through studying the fragmentation pattern using MS/MS spectra. In addition, the retention time (Rt) served as criterion of polarity and elution order, which was used for the numbering of the compounds (Figure 1 and Table 1).

### 2.4. Isolation and Identification of Glaucolide A

As glaucolide A (1; Figure 2) was identified as one of the most representative compounds in Vp-LRE, it was isolated from the extract via chromatographic fractionation and additionally identified by ^1^H- and ^13^C-NMR data analysis in comparison to literature [24,25] ^1^H NMR (500 MHz, CDCl_3_) δ (ppm): 1.53 (3H; s, H-15); 1.64 (4H, s, H-3b and H-14); 1.92 (3H, q, J = 1.0 Hz, H-4′); 2.04 (3H, s, H-2″); 2.06 (3H, s, H-2″); 2.29 (2H, m, H-2b and H-3a); 2.59 (1H, m, H-9b); 2.79 (2H, m, H-5 and H-9a); 2.93 (1H, m, H-2a); 4.80 (3H, m, H-8, H-13a and H-13b); 4.89 (1H, d, J = 9.5 Hz, H-6); 5.69 (1H, t, J = 1.0 Hz, H-3′); 6.14 (1H, t, J = 1.0 Hz, H-3′).^13^C NMR (125 MHz, CDCl_3_) δ (ppm): 18.0 (C-4′); 18.8 (C-14); 20.7 (C-15); 21.0 (C-2‴); 22.4 (C-2″); 31.8 (C-3); 32.8 (C-2); 40.3 (C-9); 55.0 (C-13); 58.7 (C-5); 61.4 (C-4); 64.4 (C-8); 80.8 (C-6); 84.7 (C-10); 125.0 (C-11); 127.8 (C-3′); 134.7 (C-2′); 163.3 (C-7); 166.2 (C-1′); 169.7 (C-12); 170.2 (C-1‴); 170.9 (C-1″); 206.6 (C-1). The purity of the isolated glaucolide A was estimated to be higher than 95% via HPLC-DAD analysis (Figure 2).

### 2.5. HPLC-DAD Analysis of Vp-LRE and Glaucolide A

The chromatographic HPLC-DAD profile of Vp-LRE, at λ 210 nm showed the presence of six major peaks with an intense signal at the Rt of 15.04 min (Figure 2, black line). The isolated glaucolide A, analyzed at the same chromatographic conditions of Vp-LRE showed a high purity peak at Rt = 15.04 in the HPLC-DAD (Figure 2, red line).

### 2.6. In Vitro Cell Viability/Cytotoxicity Assay

RAW 264.7 cells were selected to evaluate the in vitro cell viability/cytotoxicity because macrophages play a critical role in bacterial infections, as they are responsible for engulfing and destroying invading bacteria as well as producing cytokines, chemokines and reactive oxygen species that help to eliminate bacteria and coordinate the immune response to the infection [26]. This cell line was treated with concentrations ranging from 0.5 to 30 µg/mL of Vp-LRE extract or 0.5 to 20 µg/mL of glaucolide A. After 48 h, the cell viability was determined via MTT assay. As shown in Figure 3, only on the highest concentrations, Vp-LRE (30 µg/mL) and glaucolide A (10 and 20 µg/mL) induced a significant cell viability reduction, compared to the not-treated cells. No significative difference was found between non-treated cells and cells treated with the solvent (DMSO 0.5%). As expected, the positive control (DMSO 5%) reduced the viability by around 90%.

### 2.7. In Vitro Antibacterial Activity Assessment

The results of the in vitro antibacterial activity by minimal inhibitory concentration (MIC) are presented in Table 2 and Table 3. According to these tables, the MIC values showed that Vp-LRE was effective against *Staphylococcus aureus* (ATCC 6538) and (ATCC 29213), *Escherichia coli* (ATCC 10536), *Salmonella* Choleraesuis (ATCC 10708), *Salmonella* Typhimurium (ATCC 13311) (Table 2) and MRSA 1485279, MRSA 1605677, MRSA 1664534, MRSA 1688441 and MRSA 1830466 (Table 3). Vp-LRE did not reveal antibacterial effects against the other eight bacterial strains evaluated in this study (three ATCC^®^ and five *Salmonella* clinical strains) at the highest tested concentration (>5000 µg/mL). Glaucolide A showed antibacterial activity against *S. aureus* (ATCC 6538) and *S. aureus* (ATCC 29213) (Table 2). For the other strains, no antibacterial activity was observed in the gradient tested concentration (up to 500 µg/mL).

After the establishment of MIC values (Table 2 and Table 3), MBC and the bactericidal or bacteriostatic effects were determined. Vp-LRE revealed a bacteriostatic effect, showing MIC of 5000 µg/mL against *E. coli* (ATCC 10536), *S.* Choleraesuis (ATCC 10708) and *S.* Typhimurium (ATCC 13311). In these cases, MBC was not established because 5000 µg/mL was the highest concentration used in this study, as explained before. Considering the antibacterial action of Vp-LRE under *S. aureus* (ATCC 6538) and *S. aureus* (ATCC 29213) with MIC of 625 µg/mL, the MBC values were 2500 µg/mL and 1250 µg/mL, respectively, with a bacteriostatic effect present for both strains. For MRSA 1485279, MRSA 1605677, MRSA 1664534, MRSA 1688441 and MRSA 1830466, Vp-LRE demonstrated a bacteriostatic effect and the MBC values were 2500 µg/mL against MRSA 1485279, MRSA 1664534 and MRSA 1830466, and 5000 µg/mL for MRSA 1605677 and MRSA 1688441. It is noteworthy that despite the antibacterial activity of Vp-LRE observed against *S.* Choleraesuis (ATCC 10708) (MIC = 5000 µg/mL) and *S.* Typhimurium (ATCC 13311) (MIC = 5000 µg/mL), this extract and glaucolide A were unable to inhibit the five tested clinical strains of *Salmonella*. Based on glaucolide A, it was not possible to determine the MBC values of the two reference strains of *S. aureus* (ATCC 6538 and ATCC 29213) because 500 µg/mL was the highest tested concentration.

### 2.8. Molecular Docking Interactions

Figure 4 shows the redocking of the cocrystallized ligand [[*N*-(benzyloxycarbonyl)amino]methyl]phosphate (PHOS) at the catalytic site of the β-lactamase of *S. aureus* (PDBid: 1BLH), where the RMSD value was of 1.8929 Å. Thus, it was possible to perform the molecular docking studies in the catalytic site with the proposed ligands.

Clavulanic acid, the reference compound, interacted with the catalytic site of β-lactamase through hydrogen bonding with the Ser70 residue, responsible for the nucleophilic attack on the carbonyl carbon and consequent breakage of the β-lactam ring. Hydrogen bonds with Tyr105, Asn132 and Glu166 residues, dipole–permanent interaction with Gln237 and Van der Waals interactions with the other residues were also established (Figure 5A). The free energy value was −6.6 kcal/mol (Table 4).

Figure 5 presents the conformations and main intermolecular interactions between the ligands and the catalytic site of the *S. aureus* β-lactamase. Glaucolide A (Figure 5B) established hydrogen bonding with residues Ser70, Asn170, Gln237 and Arg244, with a free energy value of −6.2 kcal/mol (Table 4). 3′,4′-Dimethoxyluteolin (Figure 5C) interacted with the Asn170 residue via hydrogen bonding, with the Asn132 residue via dipole-permanence, with the aromatic ring of Tyr105 and Tyr171 and with the Ile239 residue through π-stacking alkyl interactions and π-alkyl, yielding a free energy value of −7.1 kcal/mol (Table 4). Acacetin (Figure 5D) established hydrogen bonds with residues Ser70, Asn132 and Asn170, a π-stacking type interaction with the aromatic ring of Tyr105 and alkyl and π-alkyl interactions with Ile239 residue, and the free energy value was −7.4 kcal/mol (Table 4). The flavonoid apigenin (Figure 5E) established hydrogen bonds with residues Ser70, Ser130, Asn132 and Asn170, a π-stacking type interaction with the aromatic ring of Tyr105 and a π-alkyl interaction with Ile239 residue, and the free energy value was −7.5 kcal/mol (Table 4).

## 3. Discussion

*V. polyanthes* leaves are used in Brazilian folk medicine to treat several infectious diseases such as pneumonia, flu, coughs and uterine infections [28]. However, despite the antimicrobial potential of *V. polyanthes* leaves, a correlation between its chemical composition and its antibacterial activity has not been supported by any scientific evidence so far. Additionally, several studies have reported that flavonoids, sesquiterpene lactones and other potentially active compounds of *Vernonia* species are located in the glandular trichomes of the leaves, which are epidermal secretory structures related to the production, storage and secretion of a wide number of compounds associated with plant defense and its antimicrobial and antifeeding effects [21,22,23]. The locations of these structures on the surfaces of the plant organs allow for the extraction of these compounds by washing the leaves with organic solvents, producing a selective plant extract to obtain the stored compounds in the glandular trichomes called Vp-LRE in the present study [21,22,23].

First, after preparation of the Vp-LRE, it was analyzed via UHPLC-ESI-QTOF-MS (Figure 1), showing the presence of flavonoids and sesquiterpene lactones in this extract (mainly glaucolide A, which was the major compound as observed through HPLC-DAD analysis (Figure 2)).

Our chemical characterization of Vp-LRE is in accordance with previous studies of metabolites found in *Vernonia* species [29,30,31,32,33]. A discreet number of scientific reports, focusing on the chemical composition of *V. polyanthes*, can be found, showing the presence of mainly caffeoylquinic acids, flavonoids and sesquiterpene lactones (Table 1) [19,20,22,29,30,34]. The structures of the flavonoids acacetin, chrysoeriol, isorhamnetin, isorhamnetin isomer, kaempferide, 3′,4′-dimethoxyluteolin and 3,7-dimethoxy-5,3′,4′-trihydroxyflavone were successfully annotated, with acacetin, chrysoeriol and kaempferide being first detected in *V. polyanthes*. In addition, glaucolide A and apigenin were identified via LC/MS analysis and comparison with authentic compounds (Table 1).

The chemically characterized Vp-LRE was assayed against ATCC^®^ and clinical multidrug-resistant Gram-positive and Gram-negative bacterial strains. From the large panel of evaluated bacteria, our results clearly demonstrated the most active antibacterial action of Vp-LRE against *S. aureus* ATCC^®^ and MRSA clinical tested strains. However, in another investigation, da Costa et al. revealed that these MRSA strains might be heteroresistant (hVISA) or vancomycin intermediate *S. aureus* (VISA), which could have more resistance mechanisms than expected [35]. In this way, the antibacterial potential of *V. polyanthes* as a source of bioactive natural compounds against multidrug-resistant *S. aureus*, as observed with MRSA 1485279 (MIC = 312 µg/mL) and MRSA 1605677 (MIC = 156 µg/mL), is unquestionable. Faced with the possibility of being a hVISA or VISA, these results show that Vp-LRE can present different mode of action related to resistance beyond the presence of the mecA gene (MRSA strains).

Previously published antibacterial data of *V. polyanthes* show some different results that may be due to different methods of extraction and antimicrobial evaluation. Oliveira et al. reported a significant antimycobacterial activity of *V. polyanthes* hydroalcoholic root extract used in the treatment of respiratory diseases [11]. Jorgetto et al. described the hydroalcoholic leaf extract of this plant species bacteriostatic effect against *Bacillus cereus* (ATCC 11778), *Escherichia coli* (ATCC 8739) and *Proteus mirabilis* (ATCC 25933) at MIC of 1.8 × 10^3^, 7.2 × 10^3^ and 1.4 × 10^3^ µg/mL, respectively [18]. Silva et al. demonstrated the antibiotic effect of *V. polyanthes* methanolic extract and essential oil against *S. aureus* (MIC_90%_ of 3.3 × 10^3^ and 2.8 µg/mL, in this order) and *E. coli* (MIC_90%_ 26.9 × 10^3^ and 24.1 × 10^3^ µg/mL, respectively) clinical strains [19]. Finally, Waltrich, Hoscheid and Prochnaus evaluated this property of aqueous and methanol extracts, and hexane, dichloromethane and ethyl acetate fractions of *V. polyanthes* flowers against *E. coli* (ATCC 25922), *S. aureus* (ATCC 25923) and *P. aeruginosa* (ATCC 27853) [20]. These authors observed that the ethyl acetate fractions from aqueous and methanol extracts were active against *S. aureus* (ATCC 25923), inhibiting the bacterial growth at concentrations of 6 × 10^3^ and 12 × 10^3^ µg/disk, in this order [20]. In addition, an important aspect is the MIC values obtained with the *S. aureus* reference and clinical strains. According to the Kuete’s classification of antimicrobial activity of plant extracts [36], Vp-LRE revealed a moderate antibacterial effect (100 < MIC ≤ 625 µg/mL) against *S. aureus* (ATCC 6538), *S. aureus* (ATCC 29213), MRSA 1485279 and MRSA 1605677, contributing with the traditional uses of *V. polyanthes* in the treatment of wounds and infectious diseases caused by this microorganism [9,10,11].

Because glaucolide A was identified as the major compound in Vp-LRE, it was isolated from the extract via chromatographic fractionation, with a purity estimated as higher than 95% by HPLC-DAD data analysis and chemically identified using ^13^C- and ^1^H-NMR data analysis in comparison to literature.

The antibacterial activity of glaucolide A was investigated, showing that this sesquiterpene lactone is active against *S. aureus* strains (ATCC 6538) and (ATCC 29213), with MIC values of 250 and 500 μg/mL, respectively. However, glaucolide A showed no activity against *E*. *coli*, *S*. Choleraesuis, *S.* Typhimurium, and *P. aeruginosa* ATCC^®^ and the others MRSA and *Salmonella* strains tested at the maximum concentration gradient of 500 μg/mL. To our knowledge, glaucolide A has not been yet evaluated against these bacterial strains.

As previously reported by Picman [37], the α-methylene-γ-lactone group may not be essential for the antibacterial activities of sesquiterpene lactones. The antibacterial activity of glaucolide A, which lacks the α-methylene-γ-lactone group, was similar to vernolide, vernodaline, and vernodalol, which have this group in their structures. For instance, despite the α-methylene-γ-lactone group, vernodalol was not active against *S. aureus*, unlike glaucolide A [38,39]. Our results are in agreement with the literature, indicating that the antimicrobial activity of sesquiterpene lactones is related to many chemical characteristics, such as substituent groups, their positions and configuration on the backbone, and that the presence of the α-methylene-γ-lactone moiety or the β-substituted cyclopentenone ring residue are not mandatory for the activity [37].

Regarding cytotoxicity to mammalian cells, Vp-LRE and glaucolide A showed potential toxic effects against macrophages RAW 264.7 at the same range of antibacterial concentrations. Williams et al. [40], who investigated the cytotoxicity of the extract from the leaves of *Vernonia pachyclada* and glaucolides K, L, and M on the human ovarian cancer cell line A2780, found moderate toxic effects for the extract and these compounds, with glaucolide M being the most active. However, the hypothesis that cytotoxic effects of Vp-LRE and glaucolide A may be involved with their antibacterial actions should be further explored in the future. 

In addition, many flavonoids such as apigenin have shown direct antibacterial action, synergism with antimicrobial agents and deletion of bacterial virulence [41]. Extracts and fractions with high flavonoid content were reported to exhibit antibacterial activity [42]. In this regard, the total phenolic (2.53 ± 0.01 g GAE/100 g) and flavonoid (4.26 ± 0.04 g RE/100 g) contents quantified in the Vp-LRE was higher than those observed in other extracts of *Vernonia* species [43].

One of the strategies to solve bacterial resistance mediated by β-lactamase is the inhibition of this enzyme. Currently, three β-lactamase inhibitors are used in clinical practice in association with β-lactam antibacterial agents: clavulanic acid, tazobactam and sulbactam [44]. Our results showed that 3′,4′-dimethoxyluteolin, acacetin and apigenin (compounds identified in *V. polyanthes*) were able to interact with the catalytic site of the *S. aureus* β-lactamase with free energy values lower than of clavulanic acid (Table 4). Analyzing the chemical structures of the ligands and clavulanic acid, the oxygen atoms seem to be important for establishing hydrogen bonds with the amino acid residues of the catalytic site of the enzyme. According to this data, 3′,4′-dimethoxyluteolin, acacetin and apigenin can be promising inhibitor agents of β-lactamase found in multidrug resistant bacteria. 

In conclusion, we have reported, for the first time, the antibacterial effects of the leaf rinse extract of *V. polyanthes* (Vp-LRE) and its major compound glaucolide A against clinical multidrug-resistant Gram-positive and Gram-negative bacterial strains. Flavonoids, such as chrysoeriol, and other sesquiterpene lactones, such as piptocarphin A, were annotated in Vp-LRE after chemical characterization via UHPLC-ESI-QTOF-MS analysis. This study also provides evidence for the traditional use of *V. polyanthes* in the treatment of wounds and bacterial infectious diseases. Considering the antibacterial effects observed using Vp-LRE, although a contribution of glaucolide A may be present, we cannot discard the effects of other active compounds in the extract, such as flavonoids. Further studies are in progress to disclose other important biological effects of this medicinal plant.

## 4. Materials and Methods

### 4.1. Plant Material

*V. polyanthes* was cultivated at the Medicinal Garden of the Faculty of Pharmacy, Federal University of Juiz de Fora, Juiz de Fora city, Minas Gerais State, Southeast region of Brazil (21°46′ S, 43°22′ W), and fresh leaves were collected in January 2014. The species was identified by Dr. Fátima Regina Gonçalves Salimena and a voucher specimen (CESJ number 10329) was deposited in the Herbarium Leopoldo Krieger of this Institution. The plant name *Vernonia polyanthes* Less. has been checked with (http://www.worldfloraonline.org/search?query=Vernonia+polyanthes) (accessed on 16 November 2022) and it is a synonym of *Vernonanthura phosphorica* (Vell.) H. Rob.

### 4.2. Preparation of Plant Extract

Entire dried leaves (625 g) were rinsed with dichloromethane (5 L) via shaking for 1 min and 30 s at room temperature. The resultant solution was filtered and evaporated in a rotary evaporator at controlled temperature (35–40 °C), resulting in 14 g of the leaf rinse extract of *V. polyanthes* (Vp-LRE).

### 4.3. Total Phenolic Content Determination

The total phenolic content was determined according to the Folin–Ciocalteu colorimetric method with little adjustments using gallic acid as standard [45]. The absorbance of the resulting blue color solution was measured at 788 nm (Shimadzu^®^, UV-1800, Tokyo, Japan) after two hours at room temperature. The analyses were performed in triplicate and the results were expressed as g/100 g of gallic acid equivalents (GAE).

### 4.4. Total Flavonoid Content Determination

Aluminum chloride colorimetric method was carried out according to Sobrinho et al. with few adjustments for total flavonoid content determination using rutin as the standard [45]. Each dilution of Vp-LRE stock solution (5 mg/mL in methanol) after a semi purification was separately mixed with glacial acetic acid P.A., 20% pyridine methanolic solution, 8% aluminum chloride methanolic solution, and distilled water. After 30 min at room temperature, the absorbance of the reaction mixture was measured at 412 nm (Shimadzu^®^, UV-1800, Tokyo, Japan). The analyses were performed in triplicate and the results were expressed as g/100 g of rutin equivalents (RE).

### 4.5. Chemical Composition of Vp-LRE by Ultra-High Performance Liquid Chromatography Coupled to Quadrupole Time-of-Flight Mass Spectrometry (UHPLC/Q-TOF-MS) Analysis

The UHPLC analysis was conducted with an Acquity UHPLC system (Waters Corporation, Milford, MA, USA), supplied with a quaternary pump and an autosampler coupled to an electrospray ionization quadrupole time-of-flight tandem mass spectrometer (ESI-QTOF/MS). Separation was achieved using a BEH C_18_ column (100 mm × 2.1 mm, 1.7 μm) with a flow rate at 0.4 mL/min, and a mobile phase of water with 0.1% formic acid (A) and acetonitrile (B) with a linear gradient of 5% B to 98% B in 12 min. Before the injections (0.5 µL for Vp-LRE and 1.0 µL for the authentic standards), samples were dissolved in methanol (10 mg/mL), centrifuged and filtered in a 0.22 μm filter. ESI-MS spectrometer was operated on negative ionization mode with scan range from *m/z* 100–1000. The ion source temperature was 120 °C with a desolvation temperature of 450 °C, a capillary voltage was 2 kV, a cone voltage was 30 V, and the collision energy was in ramp mode, ranging from 15–30 eV, using nitrogen as desolvation (800 L/h) and cone gas (50 L/h). Data were centroided during acquisition, and leucine-enkephalin (*m/z* 565.2771; 200 pg/mL) (Sigma-Aldrich, Steinheim, Germany) was continuously infused as an external reference into the ESI source with automatic mass correction enabled (LockSpray™). MassLynx™ 4.1 and Chromalynx™ softwares (Waters Corp., Milford, MA, USA) were used to process all data. Moreover, along with the use of authentic standards compounds (apigenin and glaucolide A) for identification, the MS^2^ spectra were compared with literature data and online databases, such as MassBank, ChemSpider and Spectral Database for Organic Compounds. 

### 4.6. Isolation and Purification of Glaucolide A from Vp-LRE

Vp-LRE (10 g) was submitted to a vacuum liquid chromatograph system (glass columns with 3–12 cm i.d.) using silica gel (40–63 µm) as stationary phase and combinations of hexane: ethyl acetate as solvents, furnishing 6 fractions (Vp-1 to 6). Fraction Vp-5 (3.5 g, hexane- EtOAc 7:3) was subjected to column chromatography over silica gel, using hexane- EtOAc mixtures as eluent, giving seven subfractions, affording one (0.7 g, from fraction Vp-5.5). The chemical structure of compound 1 was established as glaucolide A using ^1^H- and ^13^C-NMR analysis in comparison with literature. ^1^H- and ^13^C-NMR spectra (500 MHz for ^1^H-NMR and 125 MHz for ^13^C-NMR) were registered in CDCl_3_ solutions on a Bruker 500 Advance spectrometer with trimethylsilane (TMS) as the internal standard.

### 4.7. HPLC-DAD Analysis of Vp-LRE and Glaucolide A

Vp-LRE and glaucolide A (isolated from Vp-LRE) were analyzed in a high-performance liquid chromatography (HPLC) system (Waters Corporation, Milford, MA, USA) supplied with a DAD (diode array) detector, binary pumps and an autosampler. As an analytical column, the SunFire C_18_ (5 μm particle size, 4.6 mm × 250 mm, Waters Corporation) was used. The mobile phase was water with 0.5% phosphoric acid (A) and acetonitrile (B) in gradient elution: 40–100% (B) in 0–35 min, with the flow rate of 1 mL/min and detection at 210 nm. Solutions of Vp-LRE and glaucolide A (2 mg/mL) in acetonitrile were filtered through 0.45 µm membrane filters and a volume of 30 µL was injected.

### 4.8. In Vitro Cell Viability/Cytotoxicity Assay

#### 4.8.1. Cell Line and Cell Culture Conditions

RAW 264.7 macrophage-like cell line (ATCC TIP-71; laboratory’s own stock) was cultured in RPMI-1640 supplemented with 10% fetal bovine serum (FBS), 100 U/mL of penicillin and 100 μg/mL of streptomycin at 37 °C in 5% CO_2_/95% atmosphere. At 80% confluence, these cells were subjected to the experiment or subculture.

#### 4.8.2. 3-(4,5-Dimethylthiazol-2-yl)-2,5-diphenyl Tetrazolium Bromide (MTT) Assay

The cell viability/cytotoxicity was evaluated by MTT assay [46]. RAW 264.7 cells were plated at density of 2 × 10^5^ cells/well onto 96-well flat bottom plates and incubated at 37 °C and 5% CO_2_ for 24 h. The medium was removed and the fresh medium with or without concentrations of Vp-LRE (0.5, 1, 2, 5, 10, 15 and 30 µg/mL) and glaucolide A (1, 2, 5, 10 and 20 µg/mL) were added. After 48 h (37 °C; 5% CO_2_), the supernatant was removed and 100 µL of MTT (5 mg/mL) was added to each well and incubated (37 °C; 5% CO_2_) for an additional 2 h and 30 min. Subsequently, the supernatant was removed and 100 µL/well of DMSO was added to dissolve any deposited formazan. The optical density was determined at 595 nm using a microplate reader (SpectraMax 190, Molecular Devices, Sunnyvale, CA, USA). DMSO (5%) was used as positive control. All treatments were performed in triplicate.

### 4.9. In Vitro Antibacterial Activity Assessment

#### 4.9.1. Bacterial Strains

American Type Culture Collection (ATCC^®^) bacterial strains of *Staphylococcus aureus* subsp. *aureus* ((ATCC 6538), methicillin-sensitive *Staphylococcus aureus* (MSSA)), *Staphylococcus aureus* subsp. *aureus* ((ATCC 29213), MSSA), *Escherichia coli* (ATCC 10536), *Escherichia coli* (ATCC 25922), *Salmonella enterica* subsp. *enterica* serovar Choleraesuis (ATCC 10708), *Salmonella enterica* subsp. *enterica* serovar Typhimurium (ATCC 13311), *Pseudomonas aeruginosa* (ATCC 9027) and *Pseudomonas aeruginosa* (ATCC 27853) were selected for the in vitro antibacterial activity assessment. Five methicillin-resistant *Staphylococcus aureus* (MRSA) 1485279, MRSA 1605677, MRSA 1664534, MRSA 1688441 and MRSA 1830466, two *Salmonella* spp. 1266695 and *Salmonella* spp. 1507708, and three *Salmonella* Enteritidis 1406591, *S.* Enteritidis 1418594 and *S.* Enteritidis 1428260 clinical strains isolated from patients attended at Hospital Universitário Clementino Fraga Filho, Universidade Federal do Rio de Janeiro, Rio de Janeiro, Brazil, kindly donated by MSc. Adriana Lúcia Pires Ferreira, were also tested. These strains were identified via the VITEK 2^®^ automated system (BioMérieux, Durham, NC, USA). All the bacterial strains were stored as suspensions in a 10% (*w/v*) skim milk solution containing 10% (*v*/*v*) glycerol at −20 °C before use in the bioassays. Prior to use, these strains were aerobically grown in Müeller-Hinton Agar (MHA) at 35 ± 2 °C for 18–24 h. In this article, we used *S. aureus* (ATCC 6538), *S. aureus* (ATCC 29213), *E. coli* (ATCC 10536), *E. coli* (ATCC 25922), *S.* Choleraesuis (ATCC 10708), *S.* Typhimurium (ATCC 13311), *P. aeruginosa* (ATCC 9027) and *P. aeruginosa* (ATCC 27853) to simplify the text.

#### 4.9.2. Minimal Inhibitory Concentration (MIC) Determination

The MIC values of Vp-LRE and glaucolide A (samples) and ampicillin (AMP) and chloramphenicol (CHL) (standard antibiotics) were determined using the broth microdilution method according to the M07-A9 document, with few adjustments [47]. Samples stock solutions were prepared in DMSO (solvent) and water (diluent) at 10 mg/mL (*w/v*) for Vp-LRE and 1 mg/mL for glaucolide A. Antibiotics stock solutions were prepared at 1 mg/mL with solvents and diluents recommended by CLSI [27,47]. In triplicate, two-fold serial dilutions were prepared at concentrations ranging from 5000 to 40 µg/mL (Vp-LRE) and 500 to 4 µg/mL (glaucolide A, AMP and CHL). The appropriate controls were performed. The Vp-LRE concentration gradient was established based on Fabry et al. criteria (MIC < 8000 µg/mL) [48]. Posteriorly, 10 µL of standardized bacteria suspension according to 0.5 McFarland scale were added. After incubation at 35 ± 2 °C for 16–20 h under aerobic conditions, 20 μL of 2,3,5-triphenyl tetrazolium chloride (TTC) solution (1 mg/mL) were used as an indicator of bacterial growth. The system was incubated for further 30 min, and the MIC was determined.

#### 4.9.3. Minimal Bactericidal Concentration (MBC) and Bactericidal or Bacteriostatic Effect Determinations

After determination of MIC values, MBC was established following the Andrews’ procedure by spreading of 10 µL of suspensions from wells showing no bacterial growth on MHA Petri dishes [49]. After incubation at 35 ± 2 °C for 16–20 h under aerobic conditions, MBC was determined as the lowest concentration of dilutions that prevented the visible bacterial growth after subculture on MHA Petri dishes. Bacterial growth or no bacterial growth on MHA revealed a bacteriostatic or bactericidal effect, respectively.

### 4.10. Molecular Docking

The three-dimensional structures of the ligands (glaucolide A, 3′,4′-dimethoxyluteolin, acacetin, apigenin and clavulanic acid) were drawn using the MarvinSketch 20.17 program, which were submitted for geometric optimization in the Avogadro 1.2.0 program to obtain structures that were energetically more stable. The crystallographic coordinates of the three-dimensional structure of β-lactamase from *S. aureus* were obtained from the Protein Data Bank (PDB) (PDBid: 1BLH, resolution 2.3 Å). However, the AutoDock Tools 1.5.6 program was used to prepare this enzyme, where water molecules were removed and polar hydrogens and electrical charges were added for each atom. In addition, this program was used to define the settings of the Grid Box, whose dimension was 14 × 12 × 16 points spaced 1 Å with coordinates x: 5.264, y: −7.758 and z: −6.726. The AutoDock Vina 1.1.2 program was used to perform the molecular docking, and the lowest energy conformations were selected. Then, the results were analyzed using the Discovery Studio v20.1.0.19295 2020 program [50]. The molecular docking protocol was validated by the redocking method with root-mean-square deviation (RMSD) value less than 2.0 Å.

### 4.11. Statistical Analysis

The results of the cell viability assay were given as mean values (±SD) and analyzed using GraphPad Prism (version 6.0) software. Significant differences were determined by one-way analysis of variance (ANOVA) and following Tukey’s test for multiple comparisons. The significance level in the analysis was *p* < 0.05.

## 5. Conclusions

In conclusion, we have reported, for the first time, the antibacterial effects of the leaf rinse extract of *V. polyanthes* (Vp-LRE) and its major compound glaucolide A against clinical multidrug-resistant Gram-positive and Gram-negative bacterial strains. Flavonoids, such as chrysoeriol, and other sesquiterpene lactones, such as piptocarphin A, were annotated in Vp-LRE after chemical characterization via UHPLC-ESI-QTOF-MS analysis. The undertaken study provided evidence for the traditional use of *V. polyanthes* in the treatment of wounds and bacterial infectious diseases. Considering the antibacterial effects observed by Vp-LRE, although a contribution of glaucolide A may be present, we cannot discard the effects of other active compounds in the extract, such as flavonoids. Further studies are in progress to disclose other important biological effects of this medicinal plant.

## Figures and Tables

**Figure 1 antibiotics-12-00622-f001:**
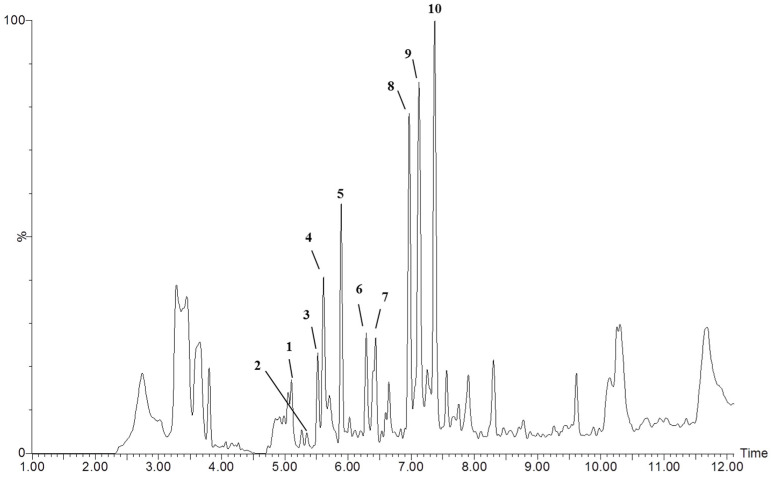
Representative UHPLC-ESI-QTOF-MS chromatogram of *V. polyanthes* leaf rinse extract (Vp-LRE) in the negative mode.

**Figure 2 antibiotics-12-00622-f002:**
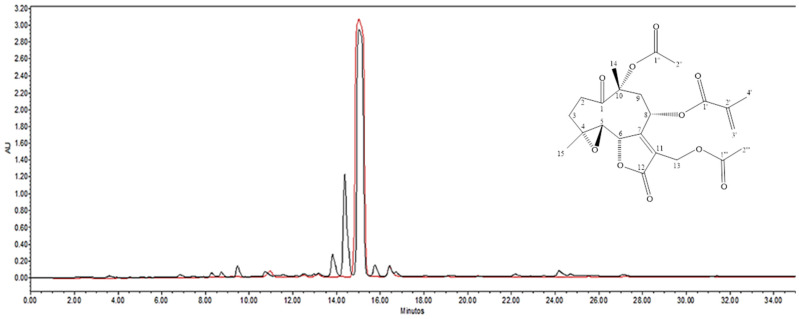
HPLC-DAD chromatograms of Vp-LRE (black line) and isolated glaucolide A (red line), and the chemical structure of glaucolide A.

**Figure 3 antibiotics-12-00622-f003:**
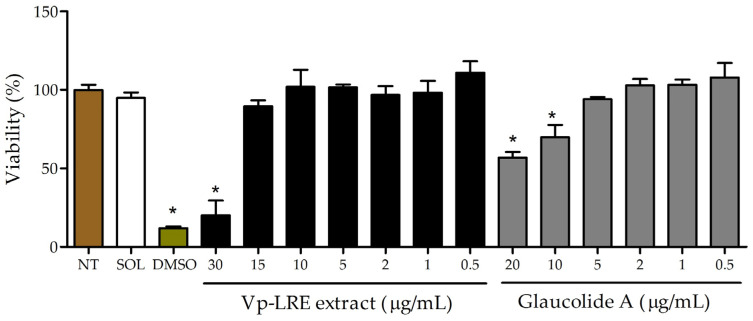
Evaluation of the cytotoxicity of *V. polyanthes* leaf rinse extract (Vp-LRE) and glaucolide A on RAW 264.7 cell line. Cell viability was evaluated via MTT assay and expressed as mean ± SD relative to untreated control (NT) (* *p* < 0.05 vs. NT).

**Figure 4 antibiotics-12-00622-f004:**
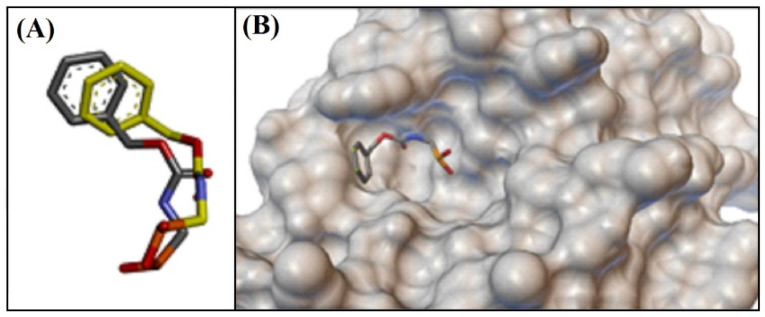
Redocking of the crystallographic ligand at the catalytic site of β-lactamase of *S. aureus*. (**A**) Crystallographic ligand conformation (moss) and conformation obtained by redocking (gray); (**B**) Ligand at the enzyme site.

**Figure 5 antibiotics-12-00622-f005:**
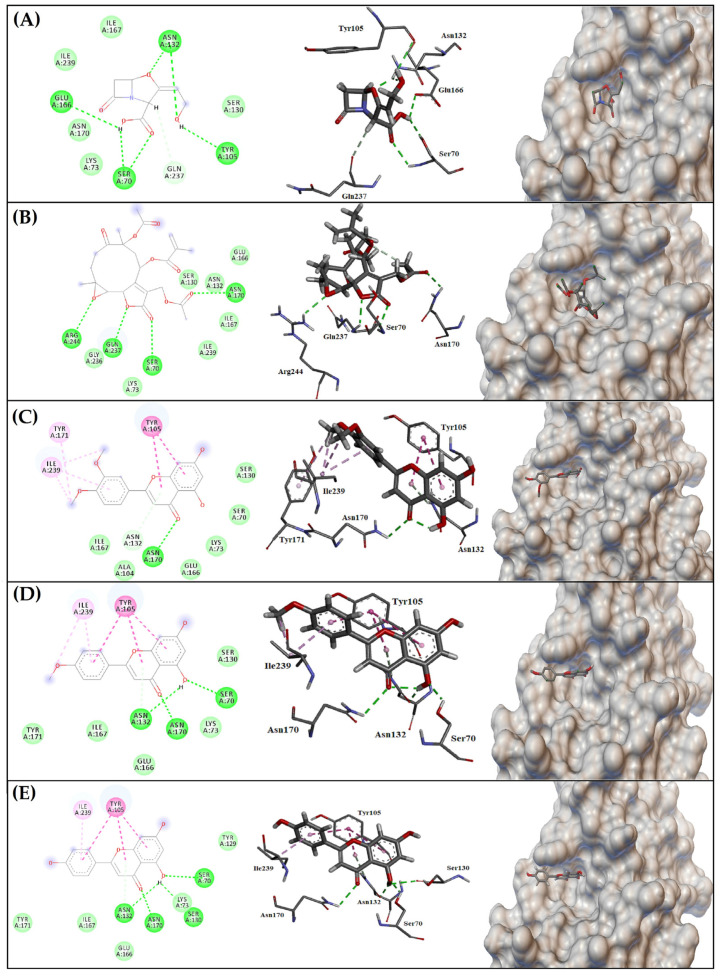
Main molecular interactions between compounds and class β-lactamase catalytic site of *S. aureus*. (**A**) Clavulanic acid. (**B**) Glaucolide A. (**C**) 3′,4′-Dimethoxyluteolin. (**D**) Acacetin. (**E**) Apigenin. Dark green: hydrogen bonding; Light green: Van der Waals; Light pink: alkyl or π-alkyl; Dark pink: π-stacking; Light blue: permanent-dipole. Dashed lines: aromatic ring.

**Table 1 antibiotics-12-00622-t001:** Chemical characterization of *V. polyanthes* leaf rinse extract (Vp-LRE) by UHPLC-MS-QTOF.

Peak	Proposed Compound	Rt * (min)	*m*/*z* ** [M − H]^−^	Main Fragments	Molecular Formula	Score	Error (ppm)
1	Isorhamnetin ^a^	5.10	315.0508	299.0956, 197.8085	C_16_H_12_O_7_	99.30	1.0
2	Apigenin ^a,b^	5.35	269.0453	197.8080, 162.8410, 116.9287	C_15_H_10_O_5_	100.00	1.1
3	Chrysoeriol	5.52	299.0562	284.0324, 256.0370, 215.1287	C_16_H_12_O_6_	99.64	1.3
4	Isorhamnetin isomer	5.61	315.0497	313.1101, 300.0261, 197.8059	C_16_H_12_O_7_	99.94	−2.5
5	3,7-dimethoxy-5,3′,4′-trihydroxyflavone ^a^	5.89	329.0670	314.0431, 299.0193, 197.8086	C_17_H_13_O_7_	99.80	2.7
6	Kaempferide	6.29	299.0549	284.0324, 197.8066, 116.9282	C_16_H_12_O_6_	99.96	−2.3
7	Piptocarphin A ^a^	6.44	457.1261 [M + Cl]^−^	441.1403, 327.1259, 197.8077, 117.9282	C_21_H_26_O_9_	99.52	−0.9
8	Acacetin	6.97	283.0622	268.0374, 239.0346, 197.8082	C_16_H_12_O_5_	99.96	8.1
9	3′,4′-dimethoxyluteolin ^a^	7.12	313.0724	298.0485, 255.0300, 197.8088	C_17_H_14_O_6_	98.26	3.8
10	Glaucolide A ^a,b^	7.37	499.1384 [M + Cl]^−^	463.1617, 403.1399, 355.1585, 275.0925, 197.8084	C_23_H_28_O_10_	99.76	2.6

* Retention time; ** Mass-to-charge Ratio; ^a^ Compounds previously reported in *Vernonia polyanthes* by Igual et al. (2013) [22]; ^b^ Confirmation with authentic standards.

**Table 2 antibiotics-12-00622-t002:** Minimal inhibitory concentration (MIC) values of *V. polyanthes* leaf rinse extract (Vp-LRE) and glaucolide A against ATCC^®^ strains.

Bacterial Strain	MIC (µg/mL)			
	Vp-LRE	Glaucolide A	AMP *	CHL *
*S. aureus* (ATCC 6538)	625	250	<4	8
*S. aureus* (ATCC 29213)	625	500	<4 ^a^	16 ^b^
*E. coli* (ATCC 10536)	5000	>500	<4	<4
*E. coli* (ATCC 25922)	>5000	>500	<4 ^d^	<4 ^d^
*S.* Choleraesuis (ATCC 10708)	5000	>500	<4	<4
*S.* Typhimurium (ATCC 13311)	5000	>500	<4	<4
*P. aeruginosa* (ATCC 9027)	>5000	>500	>500 ^c^	64 ^c^
*P. aeruginosa* (ATCC 27853)	>5000	>500	500 ^c^	64 ^c^

* MIC values of ampicillin (AMP) and chloramphenicol (CHL) were in accordance with the quality control ranges reported for nonfastidious organisms by the Clinical and Laboratory Standards Institute (CLSI), document M100-S24 [27]: ^a^ 0.5 to 2 µg/mL; ^b^ 2 to 16 µg/mL; ^c^ not reported; ^d^ 2 to 8 µg/mL. These values classify these bacteria as sensitive, with the exception of *P. aeruginosa* (ATCC 9027) and (ATCC 27853).

**Table 3 antibiotics-12-00622-t003:** Minimal inhibitory concentration (MIC) values of *V. polyanthes* leaf rinse extract (Vp-LRE) and glaucolide A against clinical strains.

Bacterial Strain	MIC (µg/mL)			
	Vp-LRE	Glaucolide A	AMP *	CHL *
MRSA 1485279	312	>500	250	64
MRSA 1605677	156	>500	250	<4
MRSA 1664534	1250	>500	16	<4
MRSA 1688441	2500	>500	250	<4
MRSA 1830466	1250	>500	64	4
*Salmonella* spp. 1266695	>5000	>500	<4	<4
*S.* Enteritidis 1406591	>5000	>500	<4	<4
*S.* Enteritidis 1418594	>5000	>500	<4	<4
*S.* Enteritidis 1428260	>5000	>500	<4	<4
*Salmonella* spp. 1507708	>5000	>500	500	<4

* MIC values of ampicillin (AMP) and chloramphenicol (CHL) were in accordance with the quality control ranges reported for nonfastidious organisms by the Clinical and Laboratory Standards Institute (CLSI), document M100-S24 [27]: These values classify these bacteria as sensitive.

**Table 4 antibiotics-12-00622-t004:** Free energy values of ligands.

Ligands	Free Energy (kcal/mol)
Clavulanic acid	−6.6
Glaucolide A	−6.2
3′,4′-Dimethoxyluteolin	−7.1
Acacetin	−7.4
Apigenin	−7.5

## Data Availability

Not applicable.

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
