# Peer review of "Vernonia polyanthes Less. (Asteraceae Bercht. & Presl), a Natural Source of Bioactive Compounds with Antibiotic Effect against Multidrug-Resistant Staphylococcus aureus"

_antibiotics, 2023, doi:10.3390/antibiotics12030622_

Round 1

Reviewer 1 Report

The authors have presented a logical design study on a relevant research topic. At present, I find no suggestions for the amelioration of the work.

Author Response

We are grateful to the reviewers for their great suggestions and comments that improved our manuscript. We carefully read the submitted version to detect possible English mistakes and to solve them. We also analyzed the reviewers' considerations one by one to make the suitable adjustments. The undertaken changes in the body text of the manuscript followed the instruction received by email (Any revisions to the manuscript should be marked up using the "Track Changes" function). Then, let us know if there is any other necessary modification to be undertaken.

REVIEWER 1

Comment: The authors have presented a logical design study on a relevant research topic. At present, I find no suggestions for the amelioration of the work.

Response:  We are grateful for the reviewer's generous comments about our study. Regarding to the quality English language (English language and style are fine/minor spell check required), we made a careful revision to adjust the possible mistakes and improve the document. As requested by reviewer, we have carefully revised the English writing and grammar to improve the comprehension as best as we can.

Reviewer 2 Report

In this submitted paper, the authors tried to investigate some biological properties of Vernonia polyanthes (V. polyanthes) leaf rinse extract (Vp-LRE) as a medicinal plant used to treat many disorders, including infectious diseases. They characterized the chemical constituents of Vp-LRE by ultra-performance liquid chromatography (UPLC/Q-TOF-MS) analysis. The authors also evaluated the cytotoxicity and antibacterial activity as well as the interactions between ligands and enzyme beta-lactamase through molecular docking. The manuscript can be accepted for publication in the Antibiotics. However, some minor issues underlined in the following comments, especially with regard to the manuscript writing language, need to be resolved before further consideration.

Minor issues:

1.      The whole manuscript writing language still contains many grammatical mistakes and errors and so the authors need to go through the paper and eliminate such critical issues.

2.      P.1, L.42, the full term has to be inserted before the abbreviation ''MRSA''.

3. P.11, Ls.374 and 383, the manufacturing city and country have to be provided for the apparatus ''Shimadzu®, UV-1800''.

Author Response

We are grateful to the reviewers for their great suggestions and comments that improved our manuscript. We carefully read the submitted version to detect possible English mistakes and to solve them. We also analyzed the reviewers' considerations one by one to make the suitable adjustments. The undertaken changes in the body text of the manuscript followed the instruction received by email (Any revisions to the manuscript should be marked up using the "Track Changes" function). Then, let us know if there is any other necessary modification to be undertaken.

REVIEWER 2

Comment: The whole manuscript writing language still contains many grammatical mistakes and errors and so the authors need to go through the paper and eliminate such critical issues.

Response: We carefully read the whole manuscript and checked the English language to eliminate grammatical mistakes and errors and improve the document. As requested by reviewer, we have carefully revised the English writing and grammar to improve the comprehension as best as we can.

Comment:  P.1, L.42, the full term has to be inserted before the abbreviation ''MRSA''.

Response: Dear Reviewer 2, the full term and abbreviation were previously inserted on page 1, line 37, and for this reason we did not repeat this description.

Comment:  P.11, Ls.374 and 383, the manufacturing city and country have to be provided for the apparatus ''Shimadzu®, UV-1800''.

Response: The manufacturing city and country were properly provided.

Reviewer 3 Report

The antimicrobial resistance in pathogenic bacteria is one of the main serious and complex public health problems of the 21st century. In this context, the effectiveness of antibiotics is declining in treating bacterial infections worldwide. In the present study, authors aimed to investigate the antibacterial activities of the leaf rinse extract of V. polyanthes (Vp-LRE) and its main compound glaucolide A against reference and clinical multidrug-resistant Gram-positive and Gram-negative bacterial strains. 

First, authors evaluated the chemical composition of (Vp-LRE) and isolated glaucolide A as it’s one of the of the most representative compounds. Then, to determine the cytotoxicity, RAW 264.7 cells were treated with various concentrations of Vp-LRE extract or glaucolide A. Finally, authors assessed the in vitro antibacterial activity of Vp-LRE extract and glaucolide A against various bacterial strains. To determine the correlation between its chemical composition and its antibacterial activity, authors used molecular docking to study potential interactions between the chemical compounds and the catalytic site of β-lactamase one of the molecular targets to solve bacterial resistance.

Overall, it’s a well-designed study with relevant and convincing results. The state of the art and research objectives were clearly presented in the introduction. The experiment was correctly designed and described in the materials and methods section and the data treatment was appropriate. 

In my opinion, the manuscript can be suitable for publication after further minor changes

Minor comments (Please see also attached file):

Abstract:  

Authors should add that is ultra-high performance liquid chromatography coupled to quadrupole time-of-flight mass spectrometry (UPLC/Q-TOF-MS).

In my opinion authors should be more cautious in their conclusion as there are many other “points” to investigate before such conclusions. May be adding a term such as potential will “solve” this issue. Also, authors should avoid use terms such as remarkable and use “significant” instead. 

Authors should avoid using “low cytotoxicity” as they only used one in vitro experiment using a murine macrophage cell line to determine cytotoxicity.

Results section: 

Authors should justify the use of RAW 264.7 murine macrophages to assess cytotoxicity. Why authors did not use human cells?

Materials and Methods section:

Authors should provide reference for RAW 264.7 cell line as they did for bacterial strains.

Author Response

We are grateful to the reviewers for their great suggestions and comments that improved our manuscript. We carefully read the submitted version to detect possible English mistakes and to solve them. We also analyzed the reviewers' considerations one by one to make the suitable adjustments. The undertaken changes in the body text of the manuscript followed the instruction received by email (Any revisions to the manuscript should be marked up using the "Track Changes" function). Then, let us know if there is any other necessary modification to be undertaken.

REVIEWER 3

Comment: Abstract: Authors should add that is ultra-high performance liquid chromatography coupled to quadrupole time-of-flight mass spectrometry (UPLC/Q-TOF-MS).

Response: We inserted the appropriate full term in the Abstract and in the Material and Methods item 4.5.

Comment: Abstract: In my opinion authors should be more cautious in their conclusion as there are many other "points" to investigate before such conclusions. May be adding a term such as potential will "solve" this issue. Also, authors should avoid use terms such as remarkable and use "significant" instead.

Response: We agree with the reviewer's comments and we inserted "potential" before "natural source" and changed "remarkable" by "significant".

Comment: Abstract: Authors should avoid using "low cytotoxicity" as they only used one in vitro experiment using a murine macrophage cell line to determine cytotoxicity.

Response: We agree and for this reason we removed "low cytotoxicity".

Comment: Results section: Authors should justify the use of RAW 264.7 murine macrophages to assess cytotoxicity. Why authors did not use human cells?

Response: Thank you for the comment. The authors decided to use RAW cell line to screen possible cytotoxicity against macrophages. In fact, macrophages play a critical role in bacterial infections, as they are responsible for engulfing and destroying invading bacteria, as well as producing cytokines, chemokines, and reactive oxygen species that help to eliminate bacteria and coordinate the immune response to the infection. Any possible cytotoxicity to macrophages would harm antibacterial effect of the tested compound, especially in further experiments. Thus, the result presented aims to correlate the activities and open perspectives for new tests such as cytokine detection. To justify the use of RAW, the authors have decided to include the following sentence to the manuscript in the 2.6 In vitro cell viability/cytotoxicity assay: "RAW 264.7 cells were selected to evaluate the in vitro cell viability/cytotoxicity because macrophages play a critical role in bacterial infections, as they are responsible for engulfing and destroying invading bacteria, as well as producing cytokines, chemokines, and reactive oxygen species that help to eliminate bacteria and coordinate the immune response to the infection.[24]"

Finally, RAW 264.7 macrophages have been used to assess cytotoxicity in comparison with antibacterial samples, as you can see in recently manuscripts published in Antibiotics Journal, such as https://doi.org/10.3390/antibiotics11111482 and https://doi.org/10.3390/antibiotics11060800.

Comment: Materials and Methods section: Authors should provide reference for RAW 264.7 cell line as they did for bacterial strains.

Response: We agree and we provided the appropriate reference (page 12, lines 454, 455).

Reviewer 4 Report

Title: Vernonia polyanthes Less. (Asteraceae Bercht. & Presl), a natural source of bioactive compounds with antibiotic effect against multidrug-resistant Staphylococcus aureus

The overall goal of this study is to determine the antibiotic effect of Vernonia polyanthes Less. (Asteraceae Bercht. & Presl), against multidrug-resistant Staphylococcus aureus. The manuscript was very well written with extensive studies done to chemically characterize V. polyanthes leaf rinse extract (Vp-LRE). Other studies performed include assessing the cytotoxicity, antibacterial activity, and molecular docking in silico experiments. Overall, the study performed is very comprehensive and will offer new insights related to the antibiotic effect of the abovementioned plant.

My major concern, however, is the novelty of the project. As stated in the introduction, previous antibacterial and other suites of studies were already performed in the medicinal plant, but it was unclear how this study will differ from previous ones. I think the novelty of the study should be emphasized in the introduction.

Author Response

We are grateful to the reviewers for their great suggestions and comments that improved our manuscript. We carefully read the submitted version to detect possible English mistakes and to solve them. We also analyzed the reviewers' considerations one by one to make the suitable adjustments. The undertaken changes in the body text of the manuscript followed the instruction received by email (Any revisions to the manuscript should be marked up using the "Track Changes" function). Then, let us know if there is any other necessary modification to be undertaken.

REVIEWER 4:

Comment: The overall goal of this study is to determine the antibiotic effect of Vernonia polyanthes Less. (Asteraceae Bercht. & Presl), against multidrug-resistant Staphylococcus aureus. The manuscript was very well written with extensive studies done to chemically characterize V. polyanthes leaf rinse extract (Vp-LRE). Other studies performed include assessing the cytotoxicity, antibacterial activity, and molecular docking in silico experiments. Overall, the study performed is very comprehensive and will offer new insights related to the antibiotic effect of the abovementioned plant.

My major concern, however, is the novelty of the project. As stated in the introduction, previous antibacterial and other suites of studies were already performed in the medicinal plant, but it was unclear how this study will differ from previous ones. I think the novelty of the study should be emphasized in the introduction.

Response: We are grateful for the supporting comments. To solve this issue, we inserted sentences (page 2, lines 64 – 66; page 2, lines 88 – 91; and page 2, line 93) to clarify the novelty of this study. We would like to explain that the three published articles about the antibacterial activity of V. polyanthes did not test multidrug-resistant clinical bacteria.

Round 2

Reviewer 4 Report

The authors have addressed my comments. Perhaps really try to strengthen the novelty to make the paper stronger.